# Validation of a Skin Calorimeter to Determine the Heat Capacity and the Thermal Resistance of the Skin

**DOI:** 10.3390/s23094391

**Published:** 2023-04-29

**Authors:** Pedro Jesús Rodríguez de Rivera, Miriam Rodríguez de Rivera, Fabiola Socorro, Manuel Rodríguez de Rivera

**Affiliations:** 1Department of Physics, University of Las Palmas de Gran Canaria, 35017 Las Palmas de Gran Canaria, Spain; pedrojrdrs@gmail.com (P.J.R.d.R.); fabiola.socorro@ulpgc.es (F.S.); 2Cardiology Service, Hospital Universitario Marqués de Valdecilla, 39008 Santander, Spain; miriam.mrdrs@gmail.com

**Keywords:** calibration, calorimetric sensor, direct calorimetry, non-differential calorimetry, thermal properties of the skin

## Abstract

In vivo determination of the skin’s thermal properties is of growing interest. Several types of sensors are being designed and tested. In this field, we have developed a skin calorimeter for the determination of the heat flow, the heat capacity and the thermal resistance of the skin. The calorimeter calibration consists of the determination of the parameters of the model we have chosen to represent the behavior of the device. This model considers the heat capacity and the thermal resistance of the skin, which depend on the case (body zone, subject, physical state, etc.) and also have a strong time dependence. Therefore, this work includes a validation study with reference materials. Finally, it is concluded that the heat capacity determined is a function of the thermal penetration depth of the measurement characteristics. In the case of high thermal conductivity materials in which the thermal penetration is nearly total, the heat capacity obtained coincides with that of the reference material sample.

## 1. Introduction

The development of sensors capable of monitoring any measurable parameter of the human body is always of interest. Moreover, if the measurement is performed in a non-invasive way, the interest is even greater. In the field of thermal measurement, temperature is of interest in all medical specialties. However, the measurement of the heat flow or the thermal properties of the skin is not common in medical applications. In previous works, we found that these measurements could be useful in the field of physical exercise [1] and skin pathology monitoring [2].

In recent years, different sensors have been developed to characterize the thermal properties of the skin [3,4]. These sensors have been used to identify physiological skin alterations and even skin cancer [5]. The principle of operation consists of producing a thermal perturbation on the skin. Then, the skin response is studied through contact or remote sensing. Each case requires the definition of an operating model according to the equations of heat transfer by conduction, convection and/or radiation, depending on the case. This model will relate the measured quantities with the thermal properties of the skin.

In calorimetry, the studied thermal processes are reproduced inside the instrument and isolated from possible external disturbances. Calibration consists of determining the relationship between the power developed in the process and the calorimetric signal provided by the detection system, which is usually based on thermoelectric sensors (thermopiles). These instruments incorporate a reference cell, so the measurement is differential. Calibration is usually performed with Joule dissipations, although reference processes and/or materials are often used [6]. In general, the uncertainty of the thermal measurement is 2 to 3%, although it depends on the characteristics of the process under study. In the case of our skin calorimeter, the measurement is non-differential, so it is necessary to incorporate the temperature and the power of the thermostat into the calibration.

We have developed a calorimeter to measure in vivo the heat flux (in mWcm^−2^), the thermal resistance (in KW^−1^) and the heat capacity (in JK^−1^) of a 2 × 2 cm^2^ skin region [7]. This calorimeter essentially consists of a thermopile placed between the skin and a programmable thermostat. The calorimeter is applied on the skin and the heat flow is transmitted by conduction from the skin to the thermostat through the measurement thermopile. The operating model is based on applying the conduction heat transfer equations to the calorimeter–skin system. Calibration is performed on a calibration base capable of experimentally simulating the behavior of the skin. The model relates two inputs with two outputs. The inputs are the power dissipated in the calibration base and in the calorimeter thermostat, and the outputs are the calorimetric signal measured by the thermopile and the temperature of the calorimeter thermostat.

In this work, we study how the model parameters vary when we change the calibration base. In this way, we complete the calibration of the instrument. Next, we will relate the heat capacity of the model with the skin heat capacity. Then, we will introduce a measurement procedure, and, finally, the results and conclusions will be presented.

## 2. Materials and Methods

### 2.1. Experimental System

The calorimeter used has been described in previous works [7]. It is a non-differential calorimeter that consists of a measurement thermopile (part a in Figure 1) placed between an aluminum plate (part b in Figure 1) and a thermostat (part c in Figure 1). The aluminum plate, in contact with the surface on which the measurement is performed, has a square surface of 4 cm^2^ and a thickness of 1 mm. The measurement thermopile (HOT20-65-F2A-1312 by Laird) provides the calorimetric signal and has a surface area of 13.2 × 13.2 mm^2^ and a thickness of 2.2 mm. The thermostat is a small aluminum block with a square surface of 14 × 14 mm^2^ and a thickness of 4 mm. Inside the thermostat, there is a heating resistor (10 Ω) and a temperature sensor (Pt100). The other side of the thermostat is in contact with a cooling system (part d in Figure 1) consisting of a Peltier element, a heatsink and a fan.

There is also a calibration base made of high-density expanded polystyrene with a copper plate that contains a temperature sensor and a heating resistor for calibration. The measurement and control system consists of a triple power supply (E3631A, by Keysight) and a data acquisition system (Data Acquisition System, 34970A with 34901, by Keysight) connected to a laptop via the GPIB/USB interface (82357B, by Keysight). A second independent source is used to power the cooling system fan. A program written in C++ controls the instrumentation, with a 0.5 s sampling period. Figure 1 shows the experimental system and a schematic of the skin calorimeter.

### 2.2. Calorimetric Model and Calibration

The measurement procedure consists of placing the calorimeter on the skin and modifying the thermostat temperature. Then, we relate the heat capacity and the thermal resistance of the skin to the calorimetric signal and the power dissipated in the thermostat to reach its programmed temperature. This method requires proposing an operating model and determining the model parameters with an appropriate calibration.

We use the “localized constants” or RC model (thermal Resistances and heat Capacities modeling), widely used in calorimetry [8]. The model consists of breaking down the calorimeter into two domains. The first one represents the region where the measurement is performed (skin) and the second one represents the calorimeter thermostat. A heat capacity (*C*_1_ and *C*_2_) and a temperature (*T*_1_ and *T*_2_) are associated with each domain. These domains are connected with each other and with the outside by thermal couplings of conductance *P*_12_, *P*_1_ and *P*_2_. The external temperatures are the ambient temperature (*T_room_*) and the cooling system temperature (*T_cold_*). Considering that in each domain, powers W_1_ and W_2_ are dissipated, the equations of the model are as follows:(1)$$\begin{array}{l} W_{1}=C_{1}\frac{dT_{1}}{dt}+P_{12}(T_{1}-T_{2})+P_{1}(T_{1}-T_{room})\\ W_{2}=C_{2}\frac{dT_{2}}{dt}+P_{12}(T_{2}-T_{1})+P_{2}(T_{2}-T_{cold})\\ y=k(T_{1}-T_{2}) \end{array}$$
where *y* is the calorimetric signal.

Measurements are always performed from an initial stationary state in which all variables are constant and their derivatives are zero. Correcting these variables to the initial steady state or baseline, we have the following equations:(2)$$\begin{array}{l} \Delta W_{1}=\frac{C_{1}}{k}\frac{d\Delta y}{dt}+\frac{P_{1}+P_{12}}{k}\Delta y+C_{1}\frac{d\Delta T_{2}}{dt}+P_{1}\Delta T_{2}\\ \Delta W_{2}=-\frac{P_{12}}{k}\Delta y+C_{2}\frac{d\Delta T_{2}}{dt}+P_{2}\Delta T_{2} \end{array}$$

From this model, we can consider the instrument as a system with two inputs (∆*W*_1_, ∆*W*_2_) and two outputs (∆*y*, ∆*T*_2_). The calibration of the sensor consists of determining the parameters of the calorimetric model given by Equation (2). The calibration is performed from a series of experimental measurements in which a known power is dissipated in the calibration base and in the thermostat. Figure 2 shows an experimental measurement in which two rectangular pulses of 300 mW (in the calibration base and in the thermostat) have been programmed for 150 s. The experimental curves of the calibration base (*W*_1_) and thermostat (*W*_2_) powers and the curves of the thermostat temperature (∆*T*_2_) and calorimetric output (∆*y*) variations are shown.

To determine the parameters, we use an iterative optimization method based on the Nelder–Mead algorithm [9,10]. The error criterion to minimize is the root mean square error (RMSE) between the experimental curves and those calculated by the model (Equation (3)):(3)$$\varepsilon =\alpha \varepsilon_{y}+\varepsilon_{T2}=\frac{\alpha }{np}\sqrt{\sum_{i=1}^{np}{\left(\Delta y_{\exp}[i]-\Delta y_{cal}[i]\right)}^{2}}+\frac{1}{np}\sqrt{\sum_{i=1}^{np}{\left(\Delta T_{2}{}_{\exp}[i]-\Delta T_{2cal}[i]\right)}^{2}}$$

In this equation, *np* is the number of points used in the fit (*np* = 1800 for the measurement shown in Figure 2) and α is the weight of the calorimetric signal error. For Δ*T*_2_ in °C and Δ*y* in volts, *α* = 100. The iterations end when the fit between the experimental and model-calculated curves are acceptable (see Figure 2b,c). Table 1 shows the parameter values of the model obtained through the identification process. The calibration accurately represents the behavior of the calorimeter according to the proposed model as a MIMO (multiple input–multiple output) system with two inputs and two outputs. This is evidenced by the excellent reconstruction of the output curves for a given input power.

From the calibration performed, it should be noted that the parameters related to the measurement thermopile (*k* and *P*_12_) and to the thermostat (*C*_2_ *y P*_2_) are invariant values of the calorimetric model. However, the parameters *C*_1_ and *P*_1_ are related to the calibration base. When the calorimeter is measuring on the skin, it is necessary to propose a method capable of determining the variation in the skin heat flux (∆*W*_1_ = ∆*W_skin_*) and the parameters *C*_1_ and *P*_1_. The objective of this technology is to determine the thermal properties of the skin, so it is important to check if we are able to relate the model-determined heat capacity (*C*_1_) to the skin heat capacity (*C_skin_*).

### 2.3. Skin Thermal Property Determination

The proposed calorimeter–skin model is similar to the one used in the calorimeter calibration, but in this case, the first domain contains a given volume of skin. The second domain is the sensor thermostat. Both domains are connected by the thermal coupling of the measurement thermopile of thermal conductance *P*_12_ and Seebeck coefficient k, parameters previously determined in the calibration (Table 1). We rewrite the model equation keeping the subscript 2 for the thermostat:(4)$$\Delta W_{skin}(t)=\frac{C_{0}+C_{skin}}{k}\frac{d\Delta y}{dt}+\frac{P_{skin}+P_{12}}{k}\Delta y+(C_{0}+C_{skin})\frac{d\Delta T_{2}}{dt}+P_{skin}\Delta T_{2}$$

This equation relates the thermostat temperature (∆*T*_2_) and the calorimetric signal (∆*y*) variations to the skin power (∆*W_skin_*) variation. To determine the thermal properties of the skin, it is necessary to produce a thermal perturbation on the skin and study its static and dynamic response. This perturbation consists of varying the thermostat temperature when the calorimeter is applied on the skin (Figure 3).

The procedure is as follows: first, the thermostat is programmed at a constant temperature. When the steady state is reached, a linear temperature variation is programmed, and the final temperature is maintained until a new steady state is reached. Figure 4 shows the experimental curves of this type of measurement: (a) thermostat power, (b) thermostat temperature and (c) calorimetric signal. In this case, the initial thermostat temperature is 26 °C, the final temperature is 36 °C and its linear variation is performed at a 4 Kmin^−1^ rate.

When the calorimeter is applied on the skin, the human body dissipates a power that we use as a reference. When the temperature of the thermostat increases, this heat power transmitted by conduction from the skin to the thermostat of the calorimeter decreases. From different measurements performed at a constant thermostat temperature, we found that this relationship is linear [11]. Therefore, we will assume that Δ*W_skin_(t)* has the same form as Δ*T*_2_(*t*), but with the opposite sign.
(5)$$\Delta W_{skin}(t)=-\Delta T_{2}\left|\frac{\Delta W_{skin}(t_{\max})}{\Delta T_{2}(t_{\max})}\right|$$

The increments Δ*W_skin_*(*t_max_*) and Δ*T*_2_(*t_max_*) are the values of these curves for the final time (*t_max_*) or steady state. For this steady state, we define an equivalent thermal resistance of the skin with Equation (6), where Δ*T*_2_(*t_max_*) is the total temperature variation of the thermostat and *P*_12_ is the thermal conductance of the measurement thermopile.
(6)$$R_{skin}=\frac{1}{P_{skin}}=\frac{1}{P_{1}}=\left|\frac{\Delta T_{2}(t_{\max})}{\Delta W_{skin}(t_{\max})}\right|-\frac{1}{P_{12}}$$

Two hypotheses are considered in the calculation process. The first one consists of considering that the power transmitted by conduction from the skin to the sensor has the same shape as the programmed thermostat temperature (Equation (5)). The second hypothesis consists of defining the thermal resistance of the skin given by Equation (6) and obtained for the final steady state. To determine these parameters (Δ*W_skin_*, *C*_1_ and *P_skin_*) we use the Nelder–Mead algorithm [9,10]. The error criterion is the RMSE (Equation (7)) between the experimental calorimetric signal (Δ*y_exp_*) and the one calculated by the model (Δ*y_cal_*).
(7)$$\varepsilon_{y}=\frac{1}{np}\sqrt{\sum_{i=1}^{np}{\left(\Delta y_{\exp}[i]-\Delta y_{cal}[i]\right)}^{2}}$$

Initially, we consider invariant values of the heat capacity and the thermal resistance of the skin. With this hypothesis, for the measurement represented in Figure 4, we obtain the following results: *C*_1_ = 5.56 JK^−1^, *R_skin_* = 1/*P*_1_ = 29.2 KW^−1^, Δ*W_skin_(t_max_*) = 263 mW, with an RMSE *ε_y_* = 27.5 µV. Two problems appear. The first one is related to the concept of measurement thermal depth: what is the volume of skin involved in the thermal perturbation produced? The second problem is how to relate the heat capacity determined to the heat capacity of the skin. These aspects are discussed in the next section.

## 3. Results and Discussion

In this section, we propose a solution to the problem of relating the heat capacity obtained by the method described in the previous section to the heat capacity of the skin involved in the measurement. The objective is to study the calorimeter’s ability to determine the heat capacities of substances. For this purpose, we performed an experimental study with inert materials that were used as a reference.

### 3.1. Determination of Heat Capacity of Inert Substances

To complete the calibration of the instrument and be able to relate the measured heat capacity to the real heat capacity, we performed an experimental study of the determination of an inert substance’s heat capacity. We considered three relevant cases. In the first case, we placed the calorimeter on a low-density expanded polystyrene (EPS) block. In the second case, we incorporated a small yellow brass cylinder, and in the third case, we incorporated a small Teflon cylinder (see Figure 5). EPS is a thermal insulator with a very low volumetric heat capacity. Brass is a material of high thermal conductivity, so it is assumed that, in this case, the thermal penetration should be total. Teflon is an insulating material with thermal properties of the same order of magnitude of skin. Table 2 shows the thermal properties of these materials and compares them with those of skin [12].

The study consisted of performing very long-duration measurements with the calorimeter applied on these materials. These measurements are like the ones shown in Figure 4, but with two differences: the stationary states were very long, and after increasing the temperature of the thermostat and reaching the stationary state, it proceeded to return to the initial temperature. The initial stationary temperature of the thermostat was 28 °C, then a linear increase was made to 34 °C with a 3 K/min rate, and when the stationary state was reached, the temperature returned to the initial value of 28 °C. Thus, in this measurement, the heat capacity was determined during the heating and cooling cycles. Figure 6 shows the experimental curves for the three cases.

With these measurements, we determined the heat capacity (*C*_1_) and the thermal resistance (*R*_1_ = 1/*P*_1_) values as a function of the time from the beginning of the temperature change. The calculation was performed on both heating and cooling areas of the curve. Figure 7 shows the results for the Teflon case. Red dots correspond to the identification of the heating curve and blue dots to the cooling. The green curve is an exponential fit of all points. The fit gives an RMSE of 0.011 JK^−1^ and a maximum deviation of 0.060 JK^−1^ for heat capacity. For thermal resistance, the RMSE is 0.074 KW^−1^ and the maximum deviation is 0.44 KW^−1^. This implies an uncertainty of 0.2% of the mean value for both magnitudes.

These results show that the volume of substance involved in the thermal perturbation (thermostat temperature change) increases exponentially with time. In the case shown in Figure 7, this increase occurs with a time constant of 13.9 min for the heat capacity and 15.6 min for the thermal resistance. For this reason, it is proposed to recalculate these properties by incorporating a new hypothesis into the calculation method. This new hypothesis consists of assuming that the heat capacity and thermal resistance increases have the same exponential variation given by Equation (8).
(8)$$\begin{array}{l} C_{1}(t)=C_{1}(0)+\Delta C_{1}\frac{1-\exp(-t/\tau )}{1-\exp(-t_{\max}/\tau )}\\ R_{1}(t)=\frac{1}{P_{1}(t)}=R_{1}(0)+\Delta R_{1}\frac{1-\exp(-t/\tau )}{1-\exp(-t_{\max}/\tau )} \end{array}$$

With this new hypothesis, the error (Equation (7)) between the calculated and experimental calorimetric curve decreases. For the full curve (40 min), the error decreases from 12.2 to 4.2 µV for Teflon, and from 6.2 to 3.1 µV for yellow brass. Figure 8 and Figure 9 show the results of heat capacity and thermal resistance as a function of time from the beginning of the temperature change. The fit between the heating and cooling curves is shown. The red curves correspond to the results of the invariant hypothesis. The green and blue curves correspond to the initial and final values for the case of the variant hypothesis (Equation (8)). As expected, the results of the invariant hypothesis are located between the initial and final values of the variant hypothesis. The heat capacity obtained is directly related to the thermal penetration depth, which we will discuss in the next section. The thermal resistance increases with heat capacity. The brass cylinder has a smaller diameter (9.5 mm) and its EPS insulation is larger (higher resistance) than the case of the Teflon cylinder, whose diameter (15.7 mm) is larger and EPS insulation is smaller (lower resistance).

### 3.2. Initial Heat Capacity Value and Measurement Thermal Depth

The thermal penetration depth is directly related to the volume of substance involved in the temperature variation caused by the thermostat of the calorimeter. Theoretically, the whole substance is affected to a greater or lesser degree depending on the distance from the source of the thermal disturbance. As the distance from the source of temperature variation increases, the resolution of the measuring system is not able to distinguish differences. Therefore, the heat capacity obtained tends to a constant value, as shown in Figure 8.

We chose yellow brass as a reference because its thermal conductivity is very high and we can assume that, at the final time, the thermal measurement depth has reached all the material, whose heat capacity is ≈2 JK^−1^. In this final situation, the heat capacity obtained is 4.81 J/K, so we can determine the initial heat capacity of the first element of the model C_0_ ≈ 2.81 J/K (see Equation (4)), which corresponds to the sensor. This consideration is consistent with the result obtained for the EPS, whose heat capacity is very low and only increases 0.08 J/K in the total time employed. With this consideration, we can indicate which are the real heat capacities at the beginning and at the end of the thermostat temperature change time for each case. For the case of Teflon, the heat capacity varies from 0.54 to 1.45 J/K, and for the case of brass, the heat capacity varies from 1.45 to 2.00 J/K. The heat capacity determined is directly related to the volume of substance involved in the temperature change. Considering the diameters of both cylinders, we have depths from 1.28 to 3.44 mm for Teflon and from 6.09 to 8.51 mm for brass.

### 3.3. Effects of the Thermostat Temperature Change Magnitude

According to the literature, this thermal depth should only depend on the thermal diffusivity of the material and on time. Gustafsson [13] proposes that the thermal depth is $\beta \sqrt{\alpha t}$, with *β* being a constant, α the thermal diffusivity and *t* the time. On the other hand, D’Esposito et al. [14] indicate that, for sinusoidal heating and cooling, the thermal depth decreases as the frequency increases. We performed an experimental study to analyze the dependence of the thermal depth on the amplitude of the temperature increase. Figure 10 shows the calorimetric signal and the programmed temperature for the case of applying the sensor on the Teflon cylinder (Figure 5). We determined the heat capacity for heating and cooling with temperature changes of 5, 7 and 9 °C. Figure 11 shows the results of the heat capacity and thermal resistance variation for these cases. Indeed, we find that these properties do not depend on the amplitude of the temperature change. However, as the temperature change increases, the higher the signal-to-noise ratio and, therefore, the higher the resolution. We observe a dispersion of the results for the same time used in the calculation. This dispersion is associated with the uncertainty of the sensor and the method used. Making an adjustment (black dotted line in Figure 11), we quantify the uncertainty at ±0.04 J/K for the heat capacity and ±0.4 K/W for the thermal resistance.

### 3.4. Skin Heat Capacity Determination

In this last section, we present the experimental results from when we applied the skin calorimeter to the volar and dorsal areas of the left wrist of a healthy 64-year-old male subject at rest, at an average ambient temperature of 20 °C. This is a very long-duration measurement, similar to that shown in Figure 6. Thus, we have two values for each time considered in the calculation, one corresponding to heating and the other to cooling. Although these properties in the human body are dynamic, in the resting situation of the subject and with a constant ambient temperature, they should not change much over the measured time. However, there are differences, which we have evaluated and associated with the uncertainty of the method. This uncertainty is ±0.1 J/K for heat capacity and ±0.7 K/W for heat resistance. Figure 12 shows the average results for the two wrist areas.

We note that the heat capacities in both zones are similar. Taking into account the mean value of the specific heat capacity of the skin (Table 2), we can deduce that, approximately, the thermal depth in the skin varies from 1 to 3 mm in the 40 min of the measurement. These depth values are similar to those obtained for Teflon for the same time. On the other hand, the thermal resistance of the volar zone is clearly lower than that of the dorsal zone. The volar area of the wrist is a suitable area to measure body temperature, since the surface temperature is closer to the core temperature due to the lesser thermal resistance.

## 4. Conclusions

The thermal insulation of the skin calorimeter allows the use of the conduction heat transport equations to adequately represent the calorimeter performance. These equations include the heat capacity and the thermal resistance of the skin.

Calibration is achieved by dissipating known powers and using inert substances with known thermal properties. It has been found that the heat capacity determined by the device is directly dependent on the thermal measurement depth, which has exponential time dependence.With inert substances, the uncertainty in the measurement of heat capacities is ±0.04 JK^−1^, and the uncertainty in the measurement of thermal resistance is ±0.4 KW^−1^. However, in measurements made in the dorsal and volar areas of the wrist, this uncertainty increases to ±0.1 JK^−1^ for heat capacity and ±0.7 KW^−1^ for thermal resistance. This increase is likely caused by the inherent variability of the skin, a living tissue.The skin calorimeter can be used to monitor the thermal properties of the skin. However, if periodic variations in the thermostat temperature are used, it must be taken into account that the depth of the measurement will increase with the period. For a clinical validation of the instrument, it would be necessary to perform more measurements in humans that are well designed in terms of measurement time, type of activity and temperature scheduling.

## Figures and Tables

**Figure 1 sensors-23-04391-f001:**
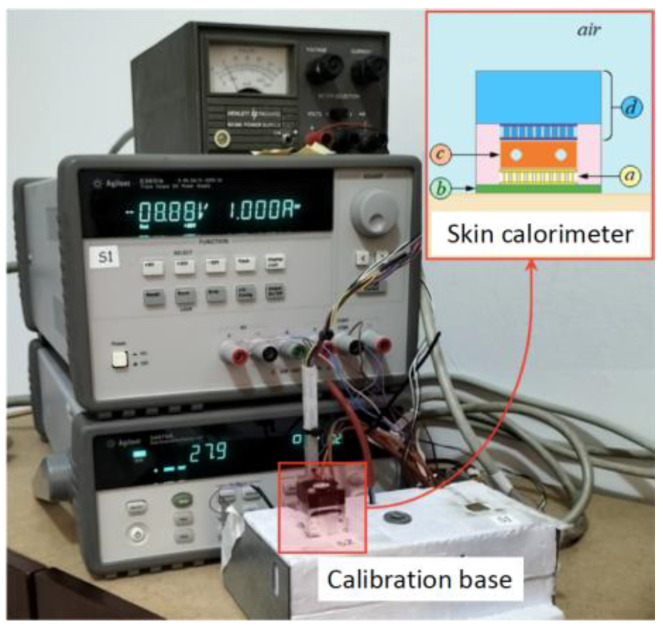
Instrumentation and skin calorimeter placed on calibration base. Schematic of the calorimeter: (a) measurement thermopile, (b) aluminum plate, (c) thermostat, (d) cooling system.

**Figure 2 sensors-23-04391-f002:**
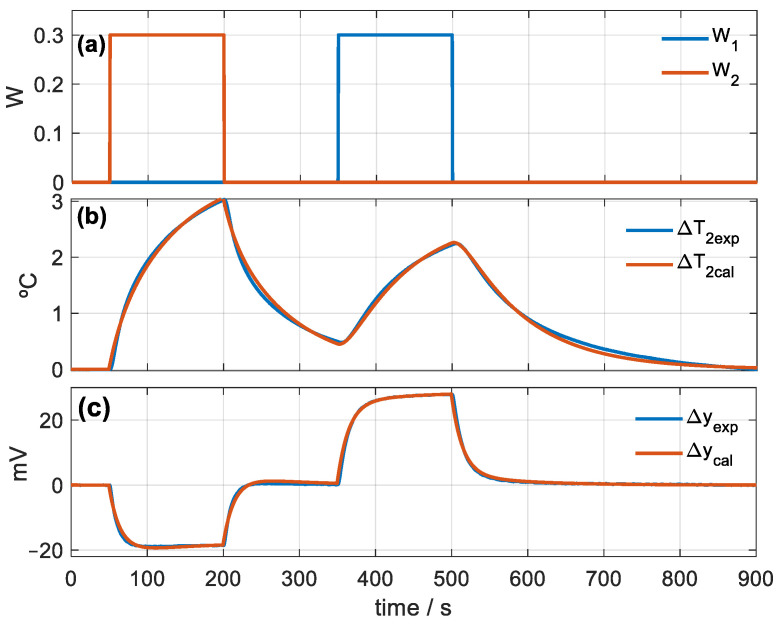
Skin calorimeter calibration measurement: (**a**) powers of the calibration base (W_1_ in blue) and of the thermostat (W_2_ in red), (**b**) variation in the thermostat temperature, (**c**) variation in the calorimetric signal. Experimental curves in blue and calculated in red.

**Figure 3 sensors-23-04391-f003:**
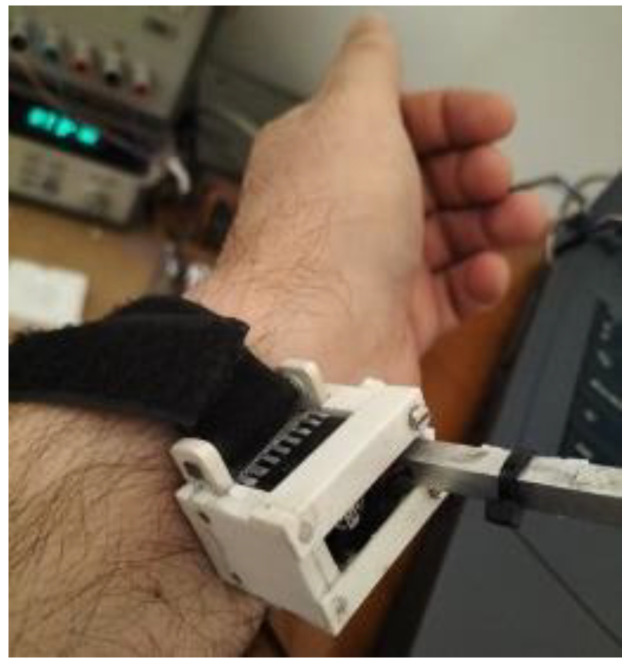
Application of the calorimeter on the wrist volar zone.

**Figure 4 sensors-23-04391-f004:**
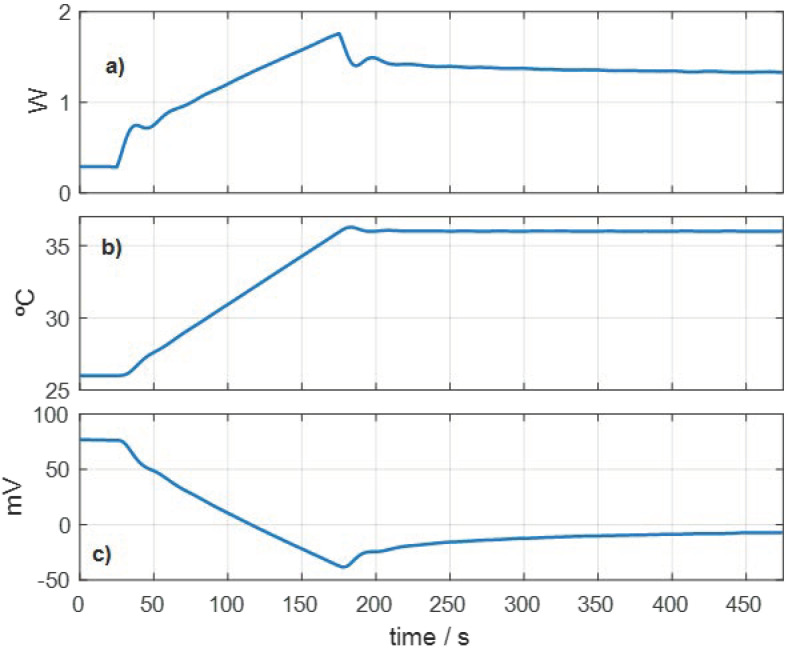
Measurement to determine the thermal properties of the skin, performed on the volar area of the right wrist of a healthy 64-year-old male subject. (**a**) Thermostat power (*W*_2_), (**b**) thermostat temperature (*T*_2_), (**c**) calorimetric signal (*y*), *T_room_* = 24.1 °C.

**Figure 5 sensors-23-04391-f005:**
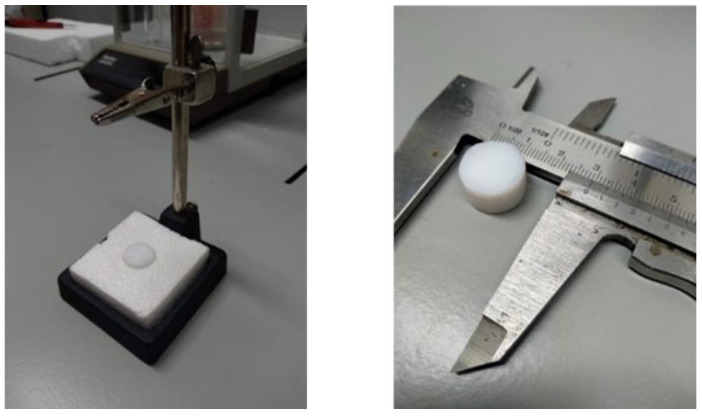
Small Teflon cylinder placed on a measurement base.

**Figure 6 sensors-23-04391-f006:**
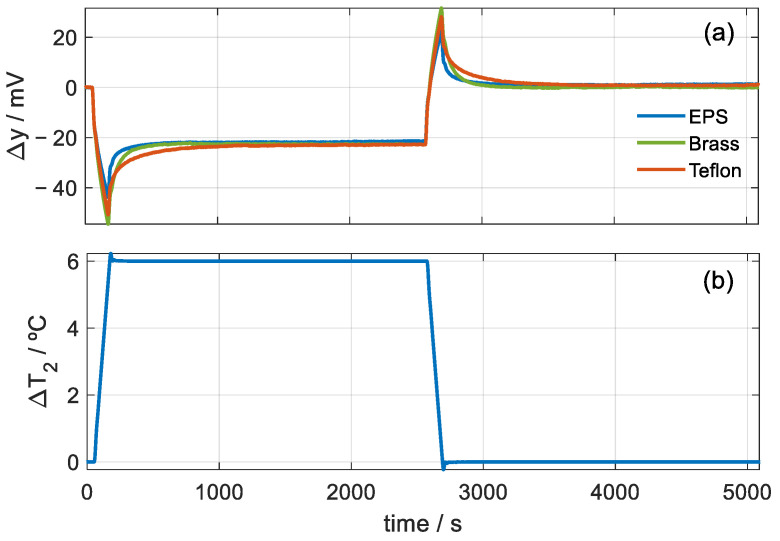
Measurements to determine the heat capacity: (**a**) variation in the calorimetric signal for EPS, brass and Teflon, (**b**) variation in the thermostat temperature.

**Figure 7 sensors-23-04391-f007:**
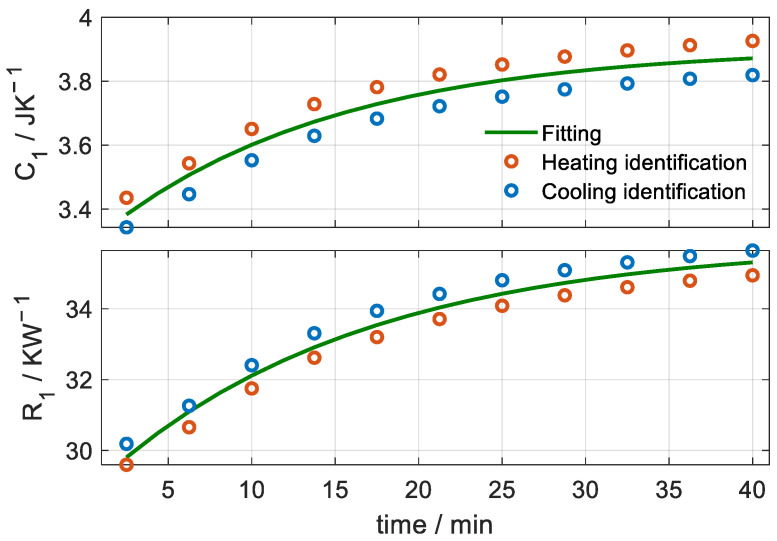
Heat capacity (*C*_1_) and thermal resistance (*R*_1_) as a function of time for the case of applying the calorimeter on a Teflon cylinder. Red dots correspond to the identification of the heating curve and blue dots to the cooling one. The green curve is a fit of all points. The results were obtained under the invariant hypothesis for each time interval considered.

**Figure 8 sensors-23-04391-f008:**
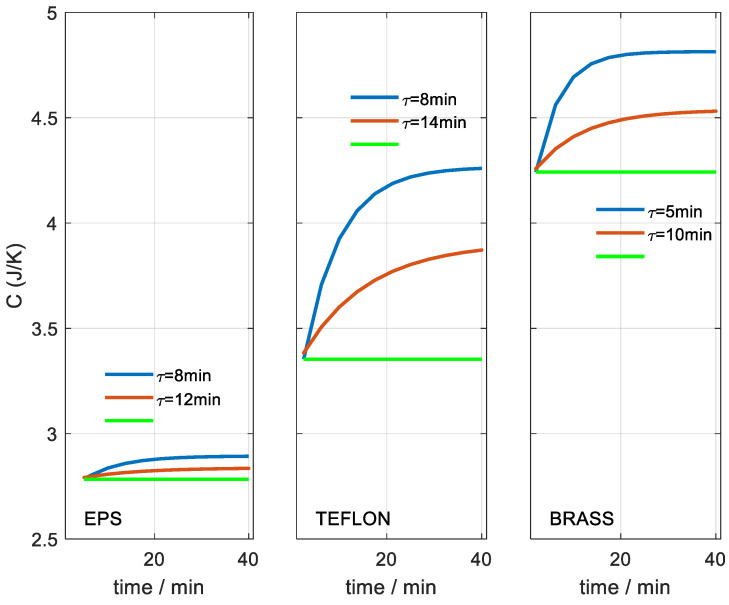
Heat capacity obtained for different materials and as a function of the time considered. Case of invariant hypothesis (red). Variant hypothesis case (Equation (7)): initial value (green) and final value (blue). The figures show the time constant of the exponential fit.

**Figure 9 sensors-23-04391-f009:**
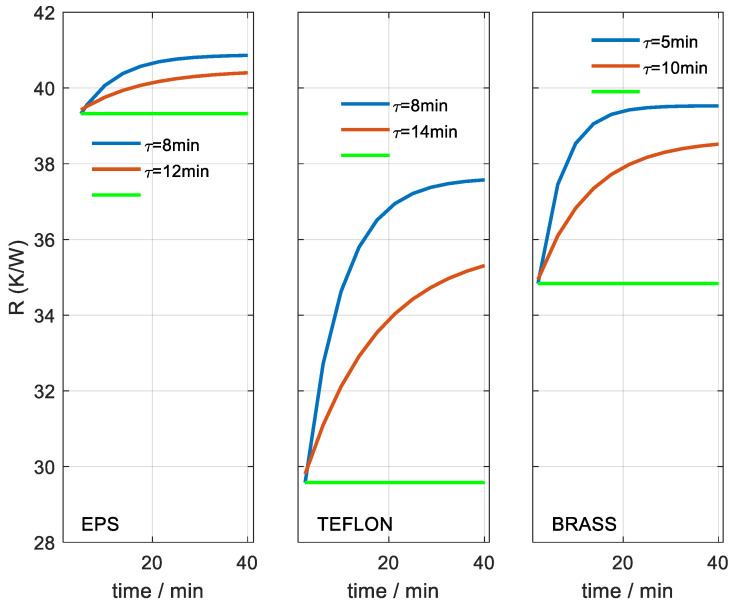
Thermal resistance obtained for different materials and as a function of the time considered. Case of invariant hypothesis (red). Variable hypothesis case (Equation (7)): initial value (green) and final value (blue). The figures show the time constant of the exponential fit.

**Figure 10 sensors-23-04391-f010:**
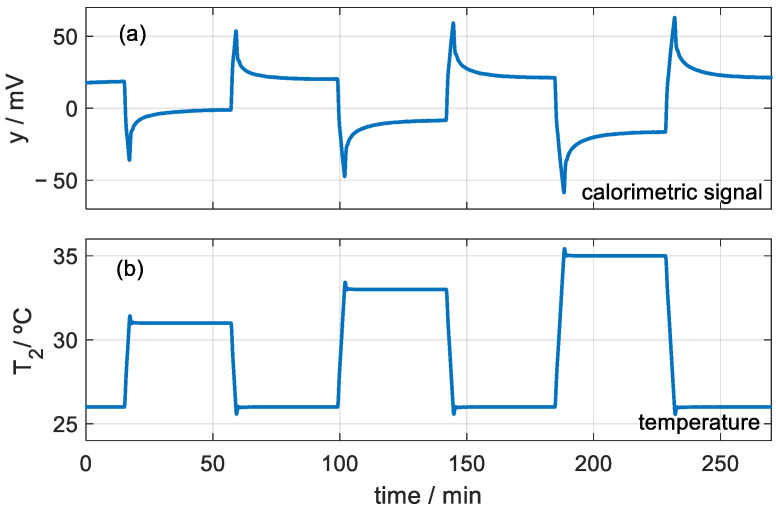
Calorimetric signal (**a**) of an experimental measurement for studying the variation in the heat capacity as a function of the thermostat temperature change (**b**).

**Figure 11 sensors-23-04391-f011:**
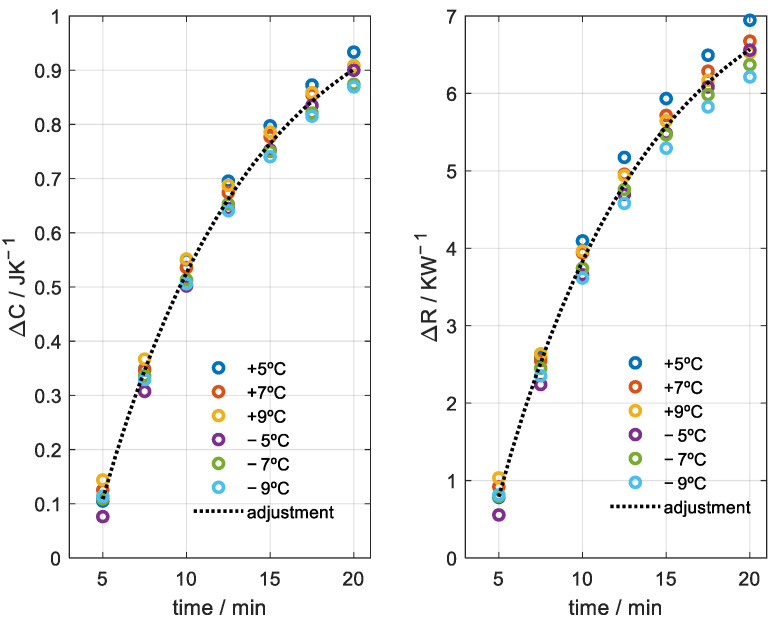
Case of Teflon cylinder at the base (see Figure 5). Variation in heat capacity and thermal resistance as a function of time for different temperature changes (+5, +7, +9, −5, −7, −9 °C). Exponential fit (dotted curve with time constant τ = 9 min).

**Figure 12 sensors-23-04391-f012:**
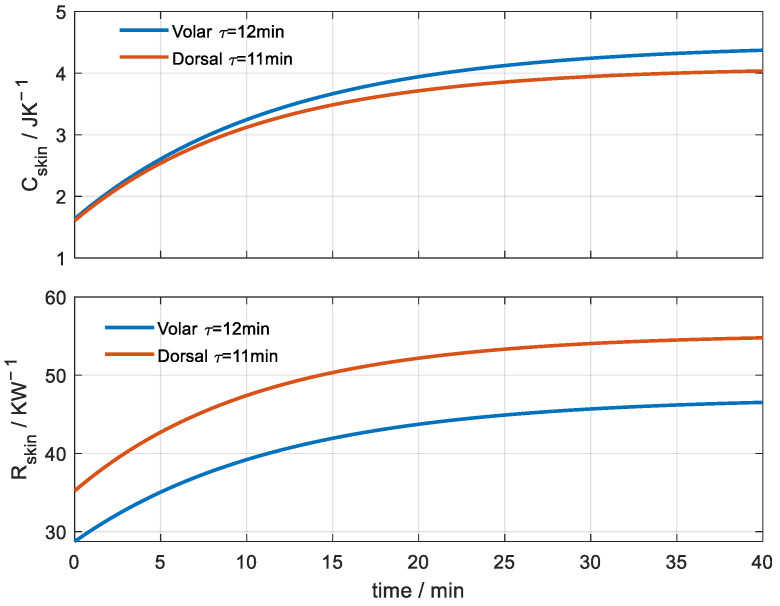
Heat capacity and thermal resistance of the volar and dorsal areas of the wrist of a healthy 64-year-old male at rest for a square area of 4 cm^2^ (T_room_ = 20 °C). The time constants of the exponential fit are shown in the figures.

**Table 1 sensors-23-04391-t001:** Calibration results when the skin calorimeter is placed on the calibration base (*C*_1_ represents the base and *C*_2_ the thermostat).

	*C*_1_ (JK^−1^)	*C*_2_ (JK^−1^)	*P*_1_ (WK^−1^)	*P*_2_ (WK^−1^)	*P*_12_ (WK^−1^)	*K* (VK^−1^)
Mean	3.617	4.373	32.52 × 10^−3^	54.45 × 10^−3^	109.65 × 10^−3^	20.82 × 10^−3^
std	0.004	0.025	0.643 × 10^−3^	1.28 × 10^−3^	0.86 × 10^−3^	0.012 × 10^−3^

RSME (Equation (3)): ε_y_ = 11.7 μV; ε_T2_ = 1.30 mK (np = 1800, Δt = 0.5 s)

**Table 2 sensors-23-04391-t002:** Thermal properties of the substances used in the experimental study of the thermal measurement depth. The reference cylinders used in the measurements have the following absolute heat capacities: *C_teflon_* = 4.2 J/K, *C_brass_* = 2.0 J/K.

Material	Thermal Diffusivity (m^2^ s^−1^)	Heat Capacity (JK^−1^ cm^−3^)	Thermal Conductivity (WK^−1^ m^−1^)
EPS	1.778 × 10^−6^	0.0225	0.040
Yellow Brass	42.636 × 10^−6^	2.5800	110.0
Teflon	0.0777 × 10^−6^	3.2186	0.250
Human Skin [12]	0.0984 × 10^−6^	3.7606	0.370

## Data Availability

Data are available on request from any of the authors.

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
