# Peer review of "Validation of a Skin Calorimeter to Determine the Heat Capacity and the Thermal Resistance of the Skin"

_sensors, 2023, doi:10.3390/s23094391_

Round 1

Reviewer 1 Report

The authors presented a skin calorimeter to determine the heat capacity and thermal resistance of skin. A model was developed to account for the heat dissipation from the measured objects to the thermostat of the calorimeter. Using samples with varied heat capacity and thermal resistance, the dependence of the heat capacity on the measurement depth was determined. The methodology of the study is sound and the presentation of the results is clear. However, the following questions need to be address to improve the quality of the paper.

1.       The introduction part can benefit from additional information about calibration of calorimeters. The authors should provide background information on what is the tolerance on the errors in terms of the measured hear capacity and heat resistance.

2.       In section 3.1, the author stated that ‘The thermal resistance increases with heat capacity and its values correspond to the base configuration’. The author summarized a correlation between the thermal resistance and heat capacity that is not generally true. I would recommend to rephase this sentence to be more specific about the measurement.

3.       The C analyzed in the paper has a unit of J/K. Why is this unit different from the unit used in the literature and mentioned in Table 2 (J/K*cm^-3)? Have the author compared the measured heat capacity of Teflon and Brass to the literature in the same unit? It would be important to elaborate on this to validate the models used in the paper.

Some minor comments:

1.       The abbreviation RC is not defined in the paper.

2.       On page 8, Figure 5 seems to be incorrectly referenced.

Overall, I recommend publication of the paper with some reservations. I would suggest discussing the limitations of the study in more details, for examples, how would the several hypotheses affect the validity of the presented calorimetry.

Author Response

Please enclosed you will find the document with detailed answers to the comments

Reviewer 2 Report

The authors developed a set of sensing devices for the measurements of heat flow, heat capacity and thermal resistance of skin, as well as thermal penetration depth. Preliminary calibrations were performed. Such a skin calorimeter may have a potential application in clinic study and fundamental research.

The paper is well organized and well written.

Author Response

Thank you very much for your comments.